# Self-Reported Computer Vision Syndrome among Thai University Students in Virtual Classrooms during the COVID-19 Pandemic: Prevalence and Associated Factors

**DOI:** 10.3390/ijerph19073996

**Published:** 2022-03-28

**Authors:** Kampanat Wangsan, Phit Upaphong, Pheerasak Assavanopakun, Ratana Sapbamrer, Wachiranun Sirikul, Amornphat Kitro, Naphasorn Sirimaharaj, Sawita Kuanprasert, Maneekarn Saenpo, Suchada Saetiao, Thitichaya Khamphichai

**Affiliations:** 1Department of Community Medicine, Faculty of Medicine, Chiang Mai University, Chiang Mai 50200, Thailand; kampanat.w@cmu.ac.th (K.W.); pheerasak.assava@cmu.ac.th (P.A.); lekratana56@yahoo.com (R.S.); wachiranun.sir@cmu.ac.th (W.S.); amornphat.kit@cmu.ac.th (A.K.); 2Department of Ophthalmology, Faculty of Medicine, Chiang Mai University, Chiang Mai 50200, Thailand; 3Faculty of Medicine, Chiang Mai University, Chiang Mai 50200, Thailand; naphasorn_siri@cmu.ac.th (N.S.); sawita_kuanpra@cmu.ac.th (S.K.); maneekarn_s@cmu.ac.th (M.S.); suchada_saetiao@cmu.ac.th (S.S.); thitichaya_kha@cmu.ac.th (T.K.)

**Keywords:** computer vison syndrome, COVID-19 impact, digital eye strain, digital screen, online study, video display terminal, visual display terminal, virtual classroom

## Abstract

During the COVID-19 pandemic, computer vision syndrome (CVS) related to online classrooms were unavoidable. This cross-sectional study aimed to explore the prevalence, characteristics and associated factors of CVS. A total of 527 students who were currently studying in a virtual classroom (70.40% female, mean (standard deviation; SD) age of 20.04 (2.17) years) were included. The prevalence of CVS assessed by an online CVS-Questionnaire was 81.0% (427/527). Comparing with those in the period before the online study, an increase in screen time (interquartile range) in students with and without CVS was 3 (0–3) and 2 (1–5) h, respectively. Overall, 516 students (97.9%) experienced at least one symptom. The most frequent symptom in CVS subjects was eye pain (96.5%). The most intense symptoms were the feeling of worsening eyesight (15.9%). The factors associated with CVS were female (*p* < 0.001), age (*p* = 0.010), atopic diseases (*p* = 0.020), prior ocular symptoms (*p* < 0.001), astigmatism (*p* = 0.033), distance from display <20 cm (*p* = 0.023), presence of glare or reflection on screen (*p* < 0.001), low screen brightness (*p* = 0.045), sleep duration (*p* = 0.030), inadequate break time between classes (*p* < 0.001) and increased screen time usage during online study (*p* < 0.001). Recommendations to prevent CVS based on the adjustable factors might reduce the burden of online study.

## 1. Introduction

The number of internet users has dramatically increased worldwide. In 2021, five billion people (65% of the world’s population) utilized the internet [1]. Rapid digital transformation during the COVID-19 pandemic escalated the number of digital display device users throughout the world. Social distancing recommendations introduced by the World Health Organization (WHO) have generated many digital innovations, including online platform business and teleconference systems [2]. According to the policy, the education sector was greatly affected [3], including Thailand. Almost all face-to-face classrooms were replaced by a virtual model.

Computer vision syndrome (CVS), caused by prolonged use of digital screen display [4], is a group of problems including ocular, visual and musculoskeletal symptoms [5]. As expected, according to increased online tasks in daily life, CVS prevalence is increasing. Many studies showed a high prevalence of CVS (60–90%) among digital display users: computer office workers and university students, including health science students [6,7,8,9,10,11]. Due to the differences in the education policy of each country, the studying time of students varies. Thai students spend over 54 h studying while Finnish, German, Swedish, Swiss and Uruguayan students spend less than 40 h studying [12]. This could affect screen time usage when changing to the online classroom. Interestingly, CVS affects the quality of life, including sleep quality and life stress [13,14], and thus a reduction in productivity and school performance. However, information regarding the prevalence and impact of CVS during the COVID-19 pandemics in Southeast Asia, including Thailand, is limited.

The associated factors of CVS are currently scarce and inconsistent. For example, Li et al. reported that itchy eye and dry eye were the most frequent symptoms of CVS and the factors associated with CVS were female gender, having refractive error and screen usage time of over 6 h (comparing with less than 2 h) [15]. Whereas, Mohan et al. reported that the most frequent symptoms were itchy eye and headache [16]. This study also showed that the factors associated with CVS were male, smart phone use and screen usage time of over 5 h [16]. To the best of our knowledge, there are currently no standard recommendations to prevent CVS. Likewise, the proposed recommendations based on the behavior of online learners, which may be different from other tasks, are limited.

This study aims to identify the prevalence, frequent symptoms and associated factors of CVS. The results could provide essential specific health information for all online learners in the future. Moreover, the results could benefit an online learning management policy for schools and universities to prevent their students from CVS.

## 2. Materials and Methods

This cross-sectional observational study was conducted during September 2021 among Chiang Mai University (CMU) students in various faculties. Those currently studying in an on-time virtual classroom were consecutively included. The sample size was calculated to be at least 386 cases using the infinite population proportion formula [17]. This calculation was based on the prevalence of CVS in a Thai population reported in previous studies (50.0–99.4%) [18,19,20]. To achieve maximum power, a prevalence of 0.5, alpha of 0.05 and error (d) of 0.05 were selected to be calculated in the formula.

The primary objective of this study was to explore the prevalence of CVS. The secondary objectives were to identify ocular/visual symptoms and the associated factors of CVS. This could support further virtual classroom protocols to reduce the incidence of CVS and decrease the burden of online learning.

An open online survey questionnaire was sent on social media platforms of a CMU student community. The survey comprised four parts: health information (gender, age, refractive error, glasses or contact lens wear, ocular diseases, eye symptoms, medication use and history of laser refractive surgery), environmental characteristics of the virtual classroom (light source and intensity, screen contrast, reflection and glare on screen), display using behavior (use/rest time and satisfaction of break time between classes) and information on CVS evaluation. Atopic diseases were defined as groups of allergic conjunctivitis, allergic rhinitis, asthma and atopic dermatitis. CVS was assessed by the CVS-Questionnaire (CVS-Q), a well-validated tool to identify the ocular health of digital display users, including eye pain, feeling that eyesight is worsening, headache, eye dryness, sensitivity to light, eye itching, excessive blinking, blurred vision, difficulty focusing for near vision, tearing, feeling of a foreign body in eyes, heavy eyelids, double vision, eye redness and colored halos around objects [21]. The original version of CVS-Q was downloaded and translated into Thai by a group of researchers and proved by an ophthalmologist before online launching. CVS was diagnosed when the sum score of CVS-Q ≥ 6.

The study was approved by the Research Ethics Committee of the Faculty of Medicine, Chiang Mai University. Online informed consent from all participants was obtained. The data from the web-based platform was downloaded and extracted into an Excel spreadsheet, and then analyzed using STATA software version 14.0 (Stata Corp., College Station, TX, USA). 

Descriptive statistics were used for narrative data. Chi-square and *t*-test were used to evaluate the association of CVS and various parameters as appropriate. A *p*-value under 0.05 was considered statistically significant. Logistic regression was performed to determine the degree of association between the variables, expressed as the odds ratio (OR) with the 95% confidence intervals (CI) and *p*-value.

## 3. Results

### 3.1. Demographic Data

The participants included 527 students, predominantly female (70.40%), with the mean (standard deviation; SD) age of 20.04 (2.17) years, from 20 faculties and 2 colleges in CMU. The prevalence of CVS was 81.0% (427/527). Table 1 compared the health profiles between the CVS and non-CVS groups. Four factors, including female (74.2% vs. 54.0%), having atopic diseases (17.8% vs. 8.0%), having prior ocular symptoms (such as dry eye, itching, red-eye and eye pain) (45.7% vs. 24.0%) and astigmatism (33.0% vs. 22%) were identified significantly more often in the CVS group. 

Overall, 516 students (97.9%) experienced at least one symptom listed in the CVS questionnaires. Focusing on the CVS group, eye pain was the most frequent symptom (96.5%), followed by a burning sensation (92.5%) and headache (90.9%) (Figure 1A). Regarding the degree of severity (Figure 1B), the most intense symptoms were the feeling of worsening eyesight (15.9%), followed by headache (14.3%) and eye pain (12.2%).

### 3.2. Characteristics of Screen Display and Usage Behavior 

Various characteristics of screen display and usage behavior between the CVS and non-CVS groups were demonstrated in the Table 2. Distance of less than 20 cm from display (52.7% vs. 40%), less brightness level (14.8% vs. 7.0%) and glare or reflection on display (47.8% vs. 29.0%) were associated with CVS. Noticeably, more than 40% of students study online with a smartphone, which has a small display screen. However, the proportion of smartphone users between the two groups was insignificant.

### 3.3. Screen Time and Rest Behaviors

Comparing between the period before and during the online study, an extended screen time (interquartile range; IQR) was greater in the CVS group (3 (0–3) vs. 2 (1–5) h) despite the lower baseline of display use (5.55 ± 2.86 vs. 6.26 ± 3.31 h) (Figure 2). Table 3 compares the screen time and rest behaviors between the CVS and non-CVS groups. In the CVS group, the average sleep duration was shorter (6.33 ± 1.17 vs. 6.63 ± 1.33 h). Likewise, the proportion with an adequate break time between classes was less (12.9% vs. 39.0%).

### 3.4. Factors Associated with CVS

The univariate analyses revealed that the factors associated with CVS were female (OR 2.45; 95% CI 1.57–3.85, *p* < 0.001), age (OR 0.86; 95% CI 0.76–0.96, *p* = 0.010), atopic diseases (OR 2.49; 95% CI 1.16–5.34, *p* = 0.020), prior ocular symptoms (OR 2.66; 95% CI 1.62–4.37, *p* < 0.001), astigmatism (OR 1.75; 95% CI 1.05–2.92, *p* = 0.033), distance from display less than 20 cm (OR 1.67; 95% CI 1.07–2.60, *p* = 0.023), presence of glare or reflection on screen (OR 2.24; 95% CI 1.40–3.59, *p* < 0.001), low screen brightness (OR 2.30; 95% CI 1.02–5.19, *p* = 0.045), sleeping duration (OR 0.82; 95% CI 0.68–0.98, *p* = 0.030), inadequate break time between classes (OR 4.32; 95% CI 2.65–7.07, *p* < 0.001) and increased screen time usage during online study (OR 1.12; 95% CI 1.06–1.20, *p* < 0.001) (Table 4).

## 4. Discussion

This study showed a high prevalence of CVS (81.0%) among Thai university students during the third wave of the COVID-19 pandemic, which markedly increased from 31.4% [22] before the pandemic. As the online or virtual classroom policy was used nationwide, our results are consistent with studies from India, Saudi Arabia and China, which were also conducted during the pandemic, in which the prevalence ranged from 73.0 to 93.6%. Compared with other reports in Thailand before the pandemic [18,19,20], the CVS prevalence in those who regularly used digital displays, such as office workers, computer workers and commercial bank workers, was also high, ranging between 50.0 and 99.4%.

The increment in screen time was suggested to be an important cause of CVS. In this study, the mean screen time was 8.58 ± 3.81 h in the CVS group versus 7.92 ± 3.32 h in non-CVS. This was increased from three hours before the pandemic, slightly higher than the other study in Thailand that reported 3.71 h [22]. In terms of screen time, our results are close to the study among university students in China during the pandemic, which ranged from 7 to 9 h [11]. 

### 4.1. Reported Symptoms

Regardless of the diagnosis of CVS, almost all (97.9%) students experienced ocular discomfort, visual disturbance or headache. The first two most frequent symptoms in our CVS subjects were eye pain and burning sensation. However, the first two of those from the previous studies were varied as follows: an open survey in India (headache and eye pain) [23], a study in children who study online in India (itching and headache) [16] and a study in undergraduate students in Saudi Arabia and Spain (headache and dry eyes) [6,24]. Headache, the third most common symptom in our study, ranked in the top two positions in the previous reports [6,16,23,24].

Regarding the severity, the most intense reported symptom in our survey was the feeling of worsening eyesight. This can be partly explained by an accommodative dysfunction after prolonged near tasks causing transient myopia. Myopia was concluded to be a factor associated with near tasks in a metanalysis [25]. Despite controversy in the association of near tasks and myopia progression, a 23-year follow-up series [26] found that fewer outdoor activities were associated with faster progression of myopia. The trend was still obvious in participants aged 20–24 years, similar to the range of age in our samples. Thus, myopic shift might be another explanation.

CVS symptoms could affect the quality of life including sleep quality and life stress [11,14], and thus a reduction in productivity and school performance. Furthermore, dry eyes—a frequent symptom of CVS in our study and many studies [6,16,23]—may affect reading/computer using and ability, impaired concentration/memory, and mental health [25]. The need for artificial tears eye drops, which are quite costly, seemed to be slightly greater in the CVS group; however, it was insignificant. As a result, these symptoms should not be ignored. A recommendation for students is to make a self-assessment regarding CVS and to manage it themselves once the symptoms occur. In case a face-to-face online lesson is required, the 20–20–20 rule (for every 20 min of display time, one may take a rest by looking 20 feet away for 20 s) is recommended to mitigate CVS [27]. 

### 4.2. Associated Factors 

According to previous literature, the factors associated with CVS comprise health profiles (gender, eye problems, sleep duration and some specific medications) [8,20,28,29,30], environmental factors (improper lighting condition, contrast and display position and presence of glare/reflection) [31,32] and display behavior (types of display, using time, break time and viewing distance) [33,34,35]. From our analysis, after adjustment for confounding factors, significant associated factors included the following.

#### 4.2.1. Gender

According to the proportion of females among Chiang Mai University students, at the year this study was conducted, females accounted for 62.17%. This could be the reason that there was female predominance (70.40%) in our study. Our results showed that females associated with CVS, consistent with the report of Bahkir et al. [23]. This could be explained in that most of the CVS symptoms were also the symptoms of dry eye syndrome of which being female (in both middle age and old age) is a risk factor [36]. The level of sex hormones was attributed to the difference between genders [36]. In contrast, Mohan et al. [16] reported that males associated with CVS. As the proportion of female in our study is much higher than males, this difference could occur by chance. Further cohort studies could confirm this controversy by an adjustment of this variable.

#### 4.2.2. Age

The results showed that increasing age was a protective factor of CVS (OR 0.86; 95% CI 0.76–0.96, *p* = 0.010) while in most previous studies, aging was a risk factor of CVS [15,16]. This can be explained by the more online-lecture classes among earlier years of university students in our population compared with those of the senior. 

#### 4.2.3. Atopic Diseases

According to our results, atopic diseases, such as allergic conjunctivitis, asthma and atopic dermatitis, were associated with CVS, consistent with the data shown in a review [37]. The explanation could be that some of the CVS symptoms overlapped those of ocular allergy and dry eye syndrome [38]. These symptoms could be triggered indirectly by antihistamine in the treatment regimen of these diseases or directly by ocular allergy, leading to tear film instability and local inflammation.

#### 4.2.4. Prior Ocular Symptoms

Dry eyes have been proposed as a contributor to CVS [7]. As symptoms of dry eyes overlap those of the CVS, students with prior ocular symptoms might have preexisted dry eye disease. This could explain the result in our study which indicated greater prior ocular symptoms including dry eye symptoms in the CVS group.

In the COVID-19 situation, patients with CVS may use artificial tears to prevent or alleviate dry eye symptoms. While self-control is essential, if the symptoms do not resolve seeing a doctor is recommended. A thorough eye examination or specific treatments for dry eye disease may be required. 

#### 4.2.5. Astigmatism

Our study found that the prevalence of refractive errors was 95.8% (myopia 64.1%, astigmatism 30.9% and hyperopia 0.75%). The prevalence of refractive errors in Thailand varies with studies [39,40]. This could be because of the different definitions of each category of refractive errors and the different subpopulations. According to a survey in secondary school children [39], the prevalence of refractive errors was 29.6% (myopia 36.1%, astigmatism 10.5% and hyperopia 1.5%). In contrast, the national survey of the Thai population aged 21–30 years old showed that the prevalence of myopia was 63.8% and hyperopia was 1.4% [40].

It is worth noting that only 44.6% of Thai students with refractive error were wearing glasses [39]. Uncorrected refractive errors or an inadequate optical compensation could cause difficulty in focusing, and eventually leading to some ocular symptoms such as eyestrain, eye discomfort and tired eye during digital display use [41,42]. Astigmatism is an important type of refractive error that cause CVS symptoms [41,42,43]. In fact, at the similar magnitude, astigmatism affects the visual acuity more than myopia and hyperopia [44]. Even an uncorrected astigmatic error of 0.50 to 1.00 diopters could lead to significant increased CVS symptoms [41,42,43]. Regarding the astigmatism subtypes, oblique astigmatism affects the visual acuity more than with-the-rule and against-the-rule astigmatism [44]. A study on the induction of an oblique astigmatism to the normal subjects [43] showed that the reading rate or reading error was not altered, but the abnormal ocular symptoms increased significantly after a 10-min reading on a digital screen. Even at the end of a near task, blurred vision when looking at the distance remained [43]. To counteract the residual astigmatism, it is suggested to overcorrect the refractive error [43]. In contact lens wearers, the use of toric contact lenses instead of the non-toric ones might reduce CVS symptoms [43]. This highlighted the importance of wearing the optimal optical compensation to minimize ocular symptoms when using displays.

#### 4.2.6. Distance from Display Less than 20 cm

The explanation of the increased CVS symptoms in those viewing distance less than 20 cm could be from the viewing distance when using smartphone is shorter than that of reading book [45]. Furthermore, comparing with other devices, the working distance for mobile phones and tablets are shorter (30 cm) while for a desktop computer it is 60 cm [46]. Therefore, the accommodative and vergence demands are increased [45]. Interestingly, a study found that the longer the reading duration, the higher the tendency of a shorter viewing distance by an individual [47]. This tendency could be a result of the induced transient myopia after prolonged reading [47]. For these reasons, devices with a longer working distance, such as a laptop or desktop computer, should be more appropriate for online study.

#### 4.2.7. Increased Screen Time

Screen time was an important factor associated with CVS. Among CVS subjects, the total screen time was 8.6 h, three hours increased from the regular activity on average. Our study also showed that each extra hour increased the 12% risk of CVS. Similarly, an Indian report during the pandemic reported an 8.7 total screen time, which was a 4.8 h increase [23]. Although lessened screen time might lower the prevalence of CVS and CVS symptoms, it is difficult to achieve practically in the current situation. Modification of other factors seems to be more feasible.

#### 4.2.8. Environmental Factors

Our study showed a significant association between CVS and reflection or glare on the screen, which is indeed adjustable. This result is consistent with the data from previous studies [32,48,49]. The improper illuminance lighting condition such as glare from the window or overhead light, reflection from the wall or ceiling, and reflection on the computer screen could also cause an enormous difference in brightness in the visual field, thus causing eye discomfort. The potential mechanism of the discomfort glare involves orbicularis oculi contraction and subsequent eye strain [34,36]. Additionally, low screen brightness, another adjustable factor, also had a significant association with CVS in our study. The suggested illuminance ratio to decrease glare discomfort is 1:3 or 3:1 between those of the task and immediate surroundings [50]. According to the OSHA, computer working area brightness should be between 20 to 50 foot-candles for cathode ray tube (CRT) displays and up to 73-foot candles for liquid crystal display (LCD) [51]. These recommendations should be implemented to create awareness among students to prevent CVS during online classes. Although improper screen and eye level seemed to be more prominent in CVS according to our results, it was not statistically significant. However, a report [52] showed that viewing angles of 30–50° contributed to the more pronounced visual symptom severity. Less than 15° of viewing angle was the optimal value that provided the lowest visual complaints. Nonetheless, there are no specific standard recommendations on the viewing distance, angle of viewing or optimal screen brightness of tablet or mobile phone. The current recommendations based on those of the OSHA [51] might be applied to all types of display. The specific recommendations regarding proper screen adjustment and usage behaviors for a tablet or mobile phone require further studies.

#### 4.2.9. Sleeping Duration

Our study found that a lower sleeping duration is associated with CVS, consistent with those summarized in a review [53]. Previous studies [28,54] indicated that quality of sleep is associated with visual fatigue. The explanation could be that asthenopia was relieved when the ciliary muscle is recovered after an adequate amount sleep time. Furthermore, Magno et al. also showed the relationship between sleep deprivation (≤6 h/day) and dry eye symptoms, which could overlap those of the CVS [54].

#### 4.2.10. Adequate Break Time between Classes

Our study showed a strong association between the self-rated inadequacy of break time between on-time lectures and CVS. This result reflected the mistake of the current timetable, which has an inadequate break time on average. If the online classroom is necessary, on-demand learning, which allows students to learn at an individual’s pace, may alleviate this problem. 

### 4.3. Limitations

First, the results of this study are based on a self-reported questionnaire. Response bias and sampling bias may occur. Physical environmental factors, including glare on screen, screen brightness, environmental brightness and differences in brightness between the display and surroundings, were based on the perception of an individual. These factors should be controlled and measured in a laboratory with valid tools to get more precise results in a future study. Second, this study focused specifically on ocular and visual problems of CVS; thus, musculoskeletal problems were not included in the survey. Third, due to the nature of the cross-sectional study, causal relationships between CVS and the associated factors could not be established. Further study with a cohort or case–control design is required to affirm these associations.

## 5. Conclusions

During the COVID-19 pandemic, social distancing was mandatory, and thus online classrooms were unavoidable. As screen time had increased, the prevalence of CVS rose among Thai university students. Several students suffer from ocular discomfort, implying that the burden of the online study was overlooked. For online study, a laptop or desktop computer is preferred compared to a mobile phone. Students are recommended to use the optimal screen brightness, adjust the screen/environmental brightness to reduce reflection/glare, and to wear optical compensation with proper correction of the refractive error. If the symptoms do not resolve, making a doctor’s appointment is advised. Furthermore, educational providers should arrange adequate break times between classes and inform the student of these recommendations to prevent CVS during classes. Finally, supporting policy should be established to help reduce the burden of a virtual classroom. 

## Figures and Tables

**Figure 1 ijerph-19-03996-f001:**
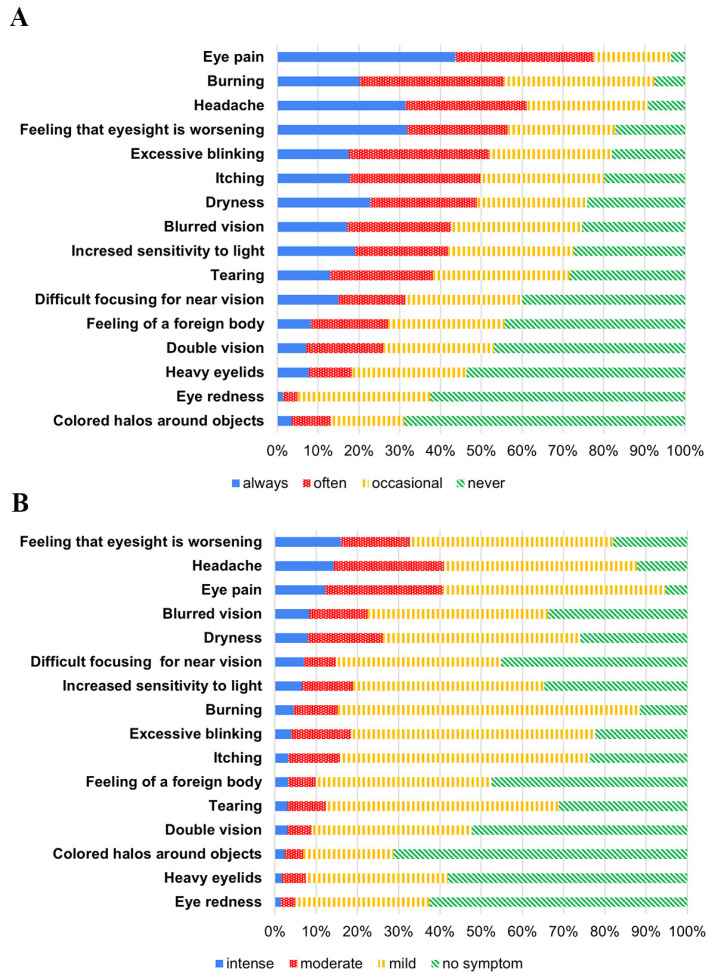
(**A**) Bar chart illustrating the frequency of ocular and visual symptoms according to CVS-Q among students with CVS. (**B**) Bar chart illustrating the severity of ocular and visual symptoms according to CVS-Q among students with CVS. Abbreviations: CVS = computer vision syndrome; CVS-Q = computer vision syndrome questionnaire.

**Figure 2 ijerph-19-03996-f002:**
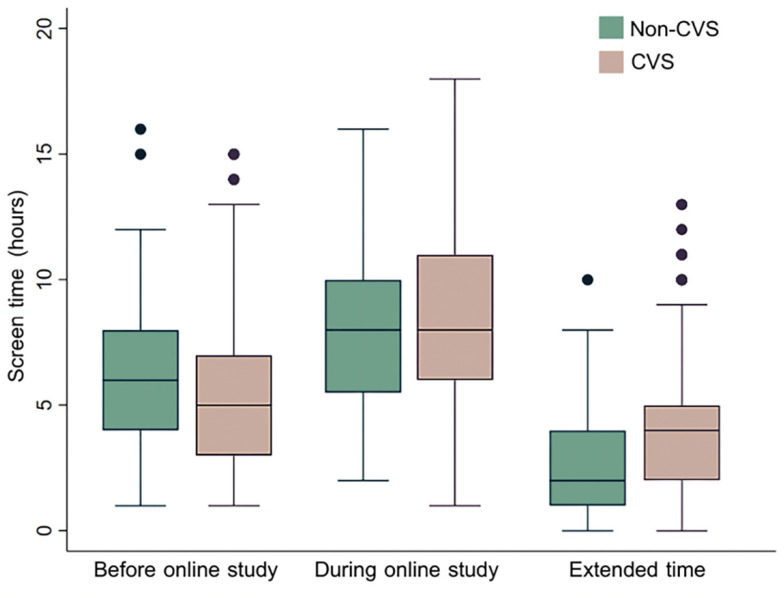
Box plots comparing screen time between the non-CVS and CVS groups: before and during online study period, and the extended time after online study period. Abbreviations: CVS = computer vision syndrome.

**Table 1 ijerph-19-03996-t001:** Demographic data.

Factors	CVS (*n* = 427)		Non-CVS (*n* = 100)		*p*-Value
Age, years (mean ± SD)	19.98 ± 1.62		20.55 ± 2.35		0.004 **
Sex, *n* (%)					
Male	110	(25.8)	46	(46.0)	<0.001 **
Female	317	(74.2)	54	(54.0)	
Atopic diseases, *n* (%)	76	(17.8)	8	(8.0)	0.015 *
Prior ocular symptoms, *n* (%)	195	(45.7)	24	(24.0)	<0.001 **
Use of dry-eye-associated systemic medications, *n* (%)	21	(4.9)	2	(2.0)	0.279
History of laser refractive surgery, *n* (%)	2	(0.5)	2	(2.0)	0.165
Myopia, *n* (%)	275	(64.4)	63	(63.0)	0.817
Hyperopia, *n* (%)	4	(0.9)	0	(0.0)	0.081
Astigmatism, *n* (%)	141	(33.0)	22	(22.0)	0.041 *
All types of refractive error, *n* (%)	295	(69.1)	65	(65.0)	0.474
Glasses wear, *n* (%)	273	(63.9)	64	(64.0)	1.000
Contact lens wear, *n* (%)	59	(13.8)	17	(17.0)	0.430
Artificial tears use, *n* (%)	74	(17.3)	11	(11.0)	0.133

* *p* < 0.05; ** *p* < 0.01. Abbreviations: CVS = computer vision syndrome.

**Table 2 ijerph-19-03996-t002:** Characteristics of screen display and usage behavior.

Variables	CVS (*n* = 427)		Non-CVS (*n* = 100)		*p*-Value
Most frequently used types of display, *n* (%)					
Tablet	208	(48.7)	57	(57.0)	0.086
Mobile phone	176	(41.2)	31	(31.0)	
Laptop	31	(7.3)	6	(6.0)	
Computer	10	(2.3)	5	(5.0)	
Projector	0	(0.0)	1	(1.0)	
Television	2	(0.5)	0	(0.0)	
Distance from display < 20 cm, *n* (%)	225	(52.7)	40	(40.0)	0.026 *
Improper display and eye level, *n* (%)	238	(55.7)	50	(50.0)	0.317
Low Screen brightness, *n* (%)	63	(14.8)	7	(7.0)	0.048 *
Glare or reflection on display, *n* (%)	204	(47.8)	29	(29.0)	0.001 **
Improper environmental brightness, *n* (%)	109	(25.5)	23	(23.0)	0.701
Difference of brightness between display and surroundings, *n* (%)	41	(9.6)	5	(5.0)	0.170
Unchanging of posture during study, *n* (%)	262	(61.4)	67	(67.0)	0.305

* *p* < 0.05; ** *p* < 0.01. Abbreviations: CVS = computer vision syndrome.

**Table 3 ijerph-19-03996-t003:** Screen time and rest behaviors.

Characteristics	CVS (*n* = 427)	Non-CVS (*n* = 100)	*p* Value
Screen time during online study, hours (mean ± SD)	8.58 ± 3.81	7.92 ± 3.32	0.102
Screen time before online study, hours (mean ± SD)	5.55 ± 2.86	6.26 ± 3.31	0.031 *
Increment of screen usage time during online study, hours (median, IQR)	3 (0–3)	2 (1–5)	<0.001 **
Rest interval during study, minutes (median, IQR)	10 (3–20)	10 (5–15)	0.372
Rest frequency during study, times (median, IQR)	2 (0–3)	2 (0–3)	0.962
Feeling adequate of break time between classes, n (%)	55 (12.9)	39 (39.0)	<0.001 **
Sleep duration per day, hours (mean ± SD)	6.33 ± 1.17	6.63 ± 1.33	0.027 *

* *p* < 0.05; ** *p* < 0.01. Abbreviations: CVS = computer vision syndrome; IQR = interquartile range.

**Table 4 ijerph-19-03996-t004:** Factors associated with computer vision syndrome.

Factors	Crude OR	95% CI	*p* Value
**Systemic factors**			
Female	2.45	1.57–3.85	<0.001 **
Age	0.86	0.76–0.96	0.010 *
Atopic diseases	2.49	1.16–5.34	0.020 *
Administration of systemic-medication related dry eye	2.53	0.58–10.99	0.214
**Ocular factors**			
Prior ocular symptoms	2.66	1.62–4.37	<0.001 **
Laser refractive surgery	0.23	0.03–1.66	0.145
Artificial tears use	1.70	0.86–3.33	0.125
Myopia	1.06	0.68–1.67	0.792
Astigmatism	1.75	1.05–2.92	0.033 *
Eyeglasses use	1.00	0.63–1.57	0.990
Contact lenses use	0.78	0.43–1.41	0.416
**Display and environment**			
Mobile phone use	1.56	0.98–2.49	0.061
Glare or reflection on screen	2.24	1.40–3.59	<0.001 **
Improper environmental brightness	1.15	0.69–1.92	0.600
Low screen brightness	2.30	1.02–5.19	0.045 *
Different brightness between display and surrounding	2.02	0.78–5.24	0.150
**Behaviors**			
Distance from screen < 20 cm	1.67	1.07–2.60	0.023 *
Proper display and eye level	1.26	0.81–1.95	0.300
Changing body posture	1.28	0.81–2.03	0.295
Duration of rest during study	1.00	0.99–1.01	0.703
Sleeping duration	0.82	0.68–0.98	0.030 *
Inadequate break time between classes	4.32	2.65–7.07	<0.001 **
Increased screen time usage during online study	1.12	1.06–1.20	<0.001 **

Abbreviations: CI = confident interval, OR = odds ratio; * *p* < 0.05; ** *p* < 0.01.

## Data Availability

Not applicable.

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
