# Peer review of "Self-Reported Computer Vision Syndrome among Thai University Students in Virtual Classrooms during the COVID-19 Pandemic: Prevalence and Associated Factors"

_ijerph, 2022, doi:10.3390/ijerph19073996_

Round 1

Reviewer 1 Report

The paper is interesting,

In the discussion, the author suggested wearing toric contact lenses in those patients with uncorreccted astigmatism error but it will depen on the method of correction of the refractive error of each patient (it should be explained)

In relation with the sample, most of the sample were female, can you explain it?

One of the most important limitation of this study is the self reported questionarie, but it has been described in the discussion.

Author Response

Dear Reviewer,

We do appreciate your valuable time reviewing our manuscript for publication with revision suggestions. Please find our attached file for the point-by-point response to the comments.

Reviewer 2 Report

This is an interesting paper about the Computer Vision Syndrome among Thai University Students in Virtual Classrooms during COVID-19 pandemic which should be improved for a its potential publication in IJERPH. Below are my comments:

I suggest being more specific in the title. For this, authors could add “Thai” or the name of the country were the study was developed.

I also suggest a more extended “Introduction” section. In my opinion, this section is poor in content. It doesn’t provide sufficient background.

On the other hand, the authors should include some discussion about other limitations of the study.  Some of physical factors are subjectively answered by the participants in the study, such as glare on the screen, brightness, or differences in brightness between display and surrounding. These factors could be controlled and measured in a laboratory and, therefore, the limitation for these factors of the on-line study should be indicated in the Discussion section. In addition, some ocular symptoms could be because of uncorrected refractive errors (or an inadequate optical compensation), which should be also commented in this section. In this sense, it should be highlighted the importance that students wear the correct optical compensation to minimize ocular symptoms when using displays. Furthermore, some information should be included about the prevalence of refractive errors in Thai (or more general, Asian) population and discuss if refractive errors reported by participants in the study are in line with prevalence reported in literature.

In addition, the authors should consider other minor comments:

- Page 2, lines 57-58. Please, justify the calculation of at least 386 cases for the sample size.

- Page 2, line 92: please, specify the term “atopic diseases”.

- Page 4, line 109: I suggest delete the repeated word “Abbreviations:”.

- Page 5, line 117. The authors indicated that the proportion of smartphone users between the two groups was insignificant. However, there is a difference between percentages of 10.2 points. I suggest that authors should be clarify this aspect in the text.

- Page 6, Figure 2. I suggest to use “Screen time (hours)” in the graph instead of “Screentimes (hours)”.

- Page 6, Table 3. Please, revise the size of numbers (row: “Rest frequency during study…”).

- Page 7, lines 160-161. Please, provide some references to support this sentence.

- Page 10, lines 261-262. Please, add a most current reference (some reference including displays of mobiles or tablets?).

Author Response

(The authors gave the same response as above.)

Reviewer 3 Report

The authors have researched the prevalence of computer vision syndrome in online classrooms during COVID-19. The number of CVS patients is expected to increase in an environment of limited access and forced use of digital devices. The results of this study, in which many students had symptoms of CVS, revealed risks other than infection with COVID-19. This study is valuable information for educators and ophthalmologists. On the other hand, I have some concerns with this paper.

Majors:

  1. What did the tool use for the online classroom? Zoom, MS Teams, or Google Meets? The on-time and on-demand lectures are different. The authors should be mention this clearly.

  1. The term health profiles are necessary to modify. What is red-eye? I think that the authors treat it as conjunctivitis. However, red-eye has some means. Also, the term worsening eyesight does not use in ophthalmology. Usually, the term visual acuity is used. The authors should be explained using the technical term.

  1. The distance is shorter using the smartphone than using the paper in the earlier study of Bababekova et al. (Optom Vis Sci, 2011). I agree with the current findings that “distance from display < 20 cm” affects CVS. The long-term and short-distance usage of smartphones is one of the risk factors of acute acquired comitant esotropia. I suggest that a subgroup analysis focusing on those who answered that their viewing distance was less than 20 cm would highlight the risk of using digital devices at close distance.

  1. In this study, the authors did not research the need for artificial tears eye drop. Although patients with CVS may use artificial tears to prevent the eyes' burning sensation, artificial tears do not improve dry eye disease. Diquafosol sodium and rebamipide eye drops are used to treat dry eye disease. While self-control is essential, if symptoms do not resolve, I recommend seeing a doctor. This is not considered an unnecessary outing. I suggest you add a sentence that the patient should go to the hospital if the limit of going out is removed.

Minors:

  1. Line 278. Typo: from adequate breaktime to adequate break time.

Author Response

(The authors gave the same response as above.)

Round 2

Reviewer 2 Report

Dear authors,

In my opinion, the manuscript has improved after revision. I've got only a couple of suggestions:

  • Lines 301 and 398. I suggest using "optical compensation" instead "optical device", since glasses or contact lenses, for example, are not considered optical devices.
  • Response to Comment 9: I suggest including the missing p-values in Table 2, so there is no need to justify in the text such difference.

Author Response

Dear Reviewer,

We do appreciate your valuable time reviewing. Please find our point-by-point responses in the attached file.
